# Using drones to improve care for HIV-exposed children in Conakry, Republic of Guinea: Anthropological perspectives

**Gabrièle Laborde-Balen** [1]*, **Oumou Hawa Diallo** [2], **Mohamed Cissé** [3‡],
**Youssouf Koita** [4‡], **Bernard Taverne** [1], **Maxime Inghels** [5], **Guillaume Breton** [6]

**1** TransVIHMI (Univ. of Montpellier, INSERM, IRD), France, **2** Solthis, Conakry, Guinea, **3** Dermatology Department, Outpatient Treatment Centre, Molecular Biology Laboratory, CHU Donka, Conakry, Guinea, **4** Programme National de Lutte contre le VIH SIDA et les Hépatites (PNLSH), Conakry, Guinea, **5** Lincoln International Institute for Rural Health, University of Lincoln, Lincoln, United Kingdom, **6** Solthis, France

☯ These authors contributed equally to this work.
‡ MC and YK also contributed equally to this work.
* gabriele.laborde-balen@ird.fr

**Data Availability Statement:** The data is sensitive because it is potentially identifying. In compliance with the confidentiality obligations recommended

## Abstract

In Conakry, Guinea, like many other African capitals, traffic congestion is a major obstacle to transporting blood samples from peripheral health centres to laboratories where tests are centralised. This situation complicates early HIV testing of HIV-exposed children (i.e., born to HIV-positive mothers), even though the World Health Organization recommends treating HIV-positive children before the age of two months to reduce mortality. The use of drones, which has proved effective in many countries for rapidly transporting healthcare products and reaching isolated areas, could help to resolve these difficulties and improve early detection. A pilot study was conducted from 2020–2021 to assess the feasibility, cost-effectiveness and acceptability of such a strategy. The pilot study had an anthropological component and this article presents the results on the acceptability and perception of using drones to optimise the transport of blood samples in Conakry. Interviews and observations were conducted across six health facilities in Conakry and in various national and international institutions, with 62 people: mothers living with HIV, health professionals and community workers, the local population, health authorities and development workers. The theoretical framework is based on the concepts of innovation and acceptability from an anthropological perspective. The analyses focus on perceptions and conditions of acceptance of a technological innovation such as drones in the healthcare sector. The results show that perceptions of drones are generally positive, despite concerns about their potential misuse. However, this consensus is fragile, knowledge on the subject is sometimes limited and public opinion can vary depending on policy changes in the political and health contexts. Future drone programmes will need to be adapted to the priority health needs identified by local stakeholders, to meet the technical and ethical challenges of this emerging technology and to develop appropriate communication to ensure an optimum level of public support.

by the National Health Research Ethics Committee of Guinea, mentioned in the consent form, it is not possible to share the full transcript of the interviews. The metadata related to this article (protocol, interview guide, book code, main socio-demographic characteristics of interviewees) are accessible on the DataSud repository (https://doi.org/10.23708/L1KYFP). Access to limited, anonymous information contained in the interview transcripts can be obtained on reasoned request by e-mail to: data@ird.fr.

**Funding:** This study is part of the pilot project "Innovative and Rapid Access to Points of Care to Optimise the Care of Newborns and Infants Exposed to HIV in Conakry, Guinea-Air POP" that is supported by ANRS MIE (Emerging infectious diseases), France; under grant number ANRS 12407 awarded to GB. The funders had no role in the study design, data collection and analysis, decision to publish, or preparation of the manuscript.

**Competing interests:** The authors have declared that no competing interests exist.

# Introduction

## Background

Despite the global progress made over the last twenty years in preventing mother-to-child HIV transmission (PMTCT), 160,000 children were born with HIV in 2021; a third of them resident in West and Central Africa, where access to PMTCT programmes is still limited [1]. The perinatal period carries a high risk of mortality. In the absence of treatment, it is estimated that half of all HIV-positive children would die before the age of two [2]. The World Health Organization (WHO) recommends early diagnosis by virological testing (Polymerase chain reaction–PCR) of children exposed to HIV at birth and after 6 weeks of life, in order to treat HIV-positive children as soon as possible [3]. These recommendations are difficult to implement in countries with limited resources because of the many difficulties that stand in the way of early detection, in particular, the lack of laboratory equipment and specialised services [4].

In Guinea, HIV prevalence amongst pregnant women is 1.5%, with significant regional disparities [5]. There are major territorial and socio-economic inequalities in women's access to healthcare. Coverage of HIV screening during pregnancy remains insufficient: only 81% of pregnant women have access to antenatal consultations (ANC) and 80% of them are offered an ANC test [6]. In total, only 64% of pregnant women are actually screened during their pregnancy. A final "catch-up" screening is recommended at the time of delivery, but with only 53% of deliveries carried out in a health facility [6], a large proportion of these women do not benefit from it. Among pregnant women screened as living with HIV, 82% receive antiretroviral treatment, but only 36% of their infants benefit from early HIV diagnosis [7]. When this screening is conducted, it often takes several months for mothers to receive the results. This means that HIV-positive children cannot be started on treatment early.

Given the lack of access to PMTCT programmes, the majority of HIV-positive children are not identified. Some children are diagnosed late, during a consultation, due to recurrent episodes of illness. In recent years, early diagnosis has been facilitated by the installation of Point of Care (POC) machines in care facilities. The WHO recommends their use, as they enable rapid results to be obtained and treatment to be prescribed on the same day, thereby reducing the risk of patients being missed [3]. But in Guinea, where HIV prevalence is low and PMTCT sites are numerous and decentralised, equipping all the country's facilities with POC platforms would not be cost-effective, given the low number of children requiring early HIV testing in each centre. The main maternity hospital in Conakry sees only around a hundred HIV-exposed children a year, while most other maternity hospitals see a few dozen.

In low-prevalence settings, in West and Central Africa, peripheral health facilities generally organise the transport of samples to a reference laboratory. This often results in long delays in obtaining results, which can take up to several months, delaying the diagnosis and treatment of HIV-positive children and consequently increasing their risk of death [4]. In the capital city of Guinea, Conakry, only a few laboratories are able to perform early HIV tests of children exposed to HIV. The government has signed an agreement with one of these laboratories, which receives most of the samples from PMTCT sites in Conakry and throughout the country. In Conakry, samples are transported by car or motorbike. The constant growth of the city and traffic congestion are considerably slowing down transport and lengthening the time it takes for samples to reach the reference laboratory.

In such circumstances, the use of drones can present an alternative or a complement to ground transport. Globally, the use of drones is booming in a variety of fields, from mapping to freight transport. They make it possible to access hard-to-reach areas quickly, cut last-mile transport costs and reduce the ecological footprint, all of which are key issues for global development [8]. The transport of healthcare products by drones is developing all over the world in

both high and low-middle income settings. In Sweden, experiments have been conducted using drones to provide emergency defibrillators [9]. Zipline, a company that has been operating in Rwanda since 2016, uses drones to transport blood bags to blood transfusion centres [10]. The Nongovernmental organization (NGO) Médecins Sans Frontières (MSF) has developed a drone transportation system for blood samples and treatments in Tanzania and Papua New Guinea [11]. In Malawi, the United Nations International Children's Emergency Fund (UNICEF) has set up a humanitarian corridor to transport medicines and blood samples using drones [12].

In Guinea, the use of civilian drones has been developing for several years, particularly in the forestry sector. However, no initiatives have used drones in urban environments. In 2020, the pilot project "Innovative and Rapid Access to Points of Care to Optimise the Care of Newborns and Infants Exposed to HIV in Conakry, Guinea" (AIR-POP), funded by the ANRS-MIE and implemented by the NGO Solthis, assessed the feasibility of transporting blood samples by drone to improve the early diagnosis of children exposed to HIV, within the city of Conakry [13]. The intervention considered in this project was to use a drone to transport blood samples as soon as they became available at the PMTCT site (i.e., "on demand") to the reference laboratory. The blood sample analysis would then be performed through a POC platform (GeneXpert) in less than two hours, and the result would be transmitted via telephone to the healthcare centre. A designated member of staff from the healthcare centre could then inform the mother of the result on the same day, or the next day, if she had returned home. The aim of this research was to test the feasibility of this intervention before it was implemented.

This feasibility study included automated test flights between a maternity unit (the maternity unit at Ignace Deen Hospital) and a laboratory located in a referral hospital (Donka University Hospital), a cost-effectiveness modelling of drone transport compared with ground transport by motorbike, and an anthropological study of the acceptability of this strategy. The success of the various automated flight tests demonstrated the feasibility of using drones in Guinean airspace in an urban environment. Modelling suggests that this strategy could be cost-effective in this context. The aim of the anthropological study, which is the focus of this article, was to analyse the perceptions and acceptability of the strategy by mothers living with HIV, healthcare providers, associations of people living with HIV, health authorities, aviation authorities and the general population.

The study took place against a backdrop of political and health instability. In January 2020, the pre-electoral period was marked by numerous demonstrations which resulted in deaths among the demonstrators. The president's re-election, contested by part of the population, was followed by a coup d'état in September 2021. On the health front, the population remains scarred by the Ebola epidemic (2014–2016), which caused the death of around 2,500 Guineans [14]. In February 2021, a new alert was brought under control by the health authorities, and in August 2021, a fatal case of Marburg fever was detected. At the same time, the country was hit by the Covid-19 pandemic, as were neighbouring countries. The first case was detected in March 2020. A second wave peaked in March 2021 and a third wave occurred in August of the same year. The epidemic affected around 36,000 people, causing more than 400 official deaths [15].

## Theoretical framework

Our theoretical approach relies on the concepts of innovation and acceptability. Sociologists and anthropologists have looked at different aspects of innovation to define its contours and explore its dynamics [16, 17]. Olivier de Sardan justifies the interest in making innovation an object of study in anthropology by the social transformations and reconfigurations it brings about [18]. Alter drew a distinction between invention and innovation [19]; for him, invention

"is only" the creation, the initial idea, whereas innovation consists of giving it meaning and putting it to use. Innovation involves a non-linear, dynamic social and organisational process–"a collective activity"–that leads to the adoption or rejection of the invention. This "*innovation process is always part of an economic logic, but its development cannot be understood without a sociological analysis of the actors who drive this process*" [19].

The role of actors is also central to 'Actor-Network Theory-ANT' or the "sociology of translation" [20, 21]. The success or failure of an innovative project depends less on the intrinsic qualities of the innovation, than on a network capable of linking together heterogeneous actors—or "actants"—both human and non-human. The relationships between them are established through an operation of "translation", whereby the actors act as "spokesmen" and translate the interests of collectives, attempting to enlist new actors. The "social" is understood as resulting from the successive interactions of these heterogeneous actors, in other words, the actor-network.

The conditions under which innovations spread have been the subject of numerous analyses. Greenhalgh et al. have explored different approaches to the dissemination and scaling-up of an innovation (mechanistic, ecological and social) and conclude that it is necessary to combine approaches for greater effectiveness, by complementing 'mechanical' efforts to reproduce an intervention with ecological and social practice perspectives [22]. In his "*diffusion theory*", Rogers identifies successive phases in the adoption of an innovation and the key elements of "*persuasion*" that determine whether it is adopted or rejected by users: the relative benefits in economic and social terms; its compatibility with the values of the group to which it belongs; its complexity or ease; the possibility of testing it; its visibility, so that others can see the results [16].

"*Trying is not adopting. And it's the adoption of an innovation that counts; on the other hand, accepting an innovation means in a sense making it one's own, appropriating it, in other words, it involves processes of identification, internalisation and interpretation*" [18]. Taking up elements of diffusion theory and combining it with other approaches, Olivier de Sardan emphasises the novel character of each innovation, which is, in itself, a unique blend, involving various strategic groups with sometimes divergent interests, making the acceptance of the innovation unpredictable. The study of the social insertion of the technical element therefore involves an exploration of the factors that determine the acceptance of the different players [23]. Louart et al. have explored the definitions and determinants of the acceptability of technological health innovations in sub-Saharan Africa. While the different types of determinants overlap with those of Rogers, the authors also emphasise the evolving nature of acceptability according to the context and stages of the intervention [24].

In this article, based on these different approaches, we explore the process by which the technical element—the drone and its applications—could become a health innovation in the specific national context of Guinea. We analyse the factors that influence the views and assessments of the various stakeholders regarding drones as objects and their potential use in transporting blood samples. In the first part, we present the stakeholders' views on the current system for transporting blood samples as part of the early diagnosis of HIV-exposed children, followed by their views on drones and their potential use to optimise early HIV diagnosis. Our discussion then turns to the perceptions of drones by the various stakeholders in the Guinean context and the conditions for the acceptability of this innovation, by contextualising our findings in line with the pertinent international scientific literature.

## Methodology

### The experimental process

The experimental process involves the dispatch of a blood sample by drone. According to WHO recommendations, the diagnosis must be made between the 4th and 6th week of life in

infants born to mothers living with HIV. A blood sample (one tube) is taken by a nurse or midwife. After the sample has been taken, a call is made to the laboratory and the drone is sent to the maternity hospital. The nurse places the sample in the appropriate device, and the drone takes it to the laboratory in just a few minutes, where it is analyzed with the POC (GeneXpert) in less than one hour. Following this, the biologist transmits the result to the caregiver via a telephone call. The caregiver then informs the mother of the result. If the test confirms the presence of HIV, the child can be treated immediately with ARVs. Routine transport is by land, and results can take several months to arrive.

The flight test was carried out between Conakry's main maternity hospital, Hôpital Ignace Deen, and the reference laboratory at the Centre Hospitalier Universitaire de Donka, where the samples are analyzed. The aircraft was supplied by Drone Volt (Villepinte, Seine-Saint-Denis, France), which has extensive experience of implementing unmanned aircraft systems in sub-Saharan Africa. It was a Hercules 2 quadricopter drone, weighing 1.8 kg and measuring 360x360 mm, with a maximum speed of 90 km/h and a range of 22 minutes. It carried two 5-ml plastic tubes (75 x 12 mm).

## Investigation dates and sites

The anthropological study took place between January 5, 2020 and June 18, 2021, in the Donka and Ignace Deen University Hospitals, and in the Kolewondy, Sonfonia, Fotoba and Saint-Gabriel Health Centers. These health facilities were involved in implementing the national programme to prevent mother-to-child transmission of HIV. They differ in size and technical facilities and are located in Conakry or its suburbs. Interviews were conducted with several government ministries concerned by the use of drones, and with various private stake-holders (international institutions, associations of people living with HIV and NGOs) involved in the health sector.

## Study population and data collection

The study population involved 62 people: 9 institutional stakeholders from the Ministry of Health, the Ministry of Defense, the Guinean Civil Aviation Authority (AGAC), the National AIDS Control Committee (CNLS) and the National AIDS and Hepatitis Control Program (PNLSH); 19 healthcare professionals (general practitioners and pediatricians, nurses, mid-wives, department heads, hospital directors, biologists, pharmacists). In some facilities (nota-bly the CHUs de Donka and Ignace Deen), the healthcare professionals were already involved in projects run by the NGO Solthis; 5 social actors or association members, including psycho-social assistants; 9 people living with HIV, including 8 pregnant women or women who had recently given birth, and the spouse of one of the women; 6 partners, members of NGOs or international organizations (Médecins sans frontières, Dream, JHPIEGO and UNAIDS); 6 people from the "general population" living near the care sites, including 2 drivers, 3 students and 1 sales manager; 8 members of the project team. We had no refusals to participate.

Data were collected via open interviews, semi-structured interviews lasting approximately one hour, and direct observation. The interview guides were developed by the senior investigator. The survey was carried out by a female senior health anthropologist (PhD) and a female Guinean medical doctor (MD) who mastered all of the relevant vernacular languages. Interviews were conducted in French or in local languages (Malinke, Soussou, Fulfulde). Interviews in local languages were translated into French. Data collection was discontinued when thematic saturation was achieved.

For this exploratory research, participants were purposively selected. The aim was to meet the primary stakeholders involved in the implementation of the national PMTCT strategy and

those involved in the development of the use of drones in the country, as well as, to gather the views of users of the healthcare system.

The themes explored were participants': (i) views on the current PMTCT system and its effectiveness; (ii) knowledge and perceptions of drones; (iii) views on the use of drones in the healthcare sector, and more specifically for transporting blood samples as part of early screening for children exposed to HIV.

Observations covered clinical consultations at each of the healthcare facilities in the study, laboratory activity, healthcare team meetings and drone deployment. They supplemented the interviews by putting comments into the perspective of practice.

The data collection was conducted in two stages: the first took place in January 2020. This allowed us to visit the project sites and carry out an initial series of interviews and observations. The second, in June 2021, took place at the same time as the flight tests. It provided an opportunity to gather feedback from participants who had attended the flights, and to complete the interviews. The comments of some interviewees were collected twice during these two periods.

## Analysis method

Interviews were recorded in multiple languages, translated and transcribed. The corpus of data, interviews and observations, was subjected to content analysis. They were manually thematically coded, according to a codebook then organized into Excel files by the senior investigator. Constant back and forth between data collection and analysis enabled us to identify a first level of categories and interactions from which the thematic guides were refined. This method is based on inductive analysis inspired by grounded theory [25]. The data were compared and put into perspective with the scientific literature.

## Ethical concerns

The study received approval from the National Health Research Ethics Committee of Guinea on December 27, 2019 (Ref: 119/CNERS/19). All participants received information about the study and were able to ask questions prior to taking part. Written consent was obtained from patients, and verbal consent was obtained from healthcare professionals. Interviews were anonymized and pseudonyms used. All study procedures complied with the 1964 Declaration of Helsinki and its subsequent amendments, as well as the ANRS-MIE ethical charter of 2002, revised in 2017. A presentation of the result was organized for authorities and healthcare professionals.

## Results

### Views on the current system

The Guinean health authorities acknowledged the weaknesses of the PMTCT programme. They were aware of the difficulties involved in reaching children exposed to HIV, obtaining rapid results, caring for HIV-positive children and including them in cohorts of children treated with ARVs. For them, the entire cascade of care for both, mother and child, remained a considerable challenge. They believed that the early diagnosis system worked better in Conakry than in regions outside of the capital, where it is virtually non-existent. The lack of POC platforms in operation, poor inter-city transport and difficulties in accessing health facilities during the rainy season were all major obstacles. According to Ministry of Health advisers, the development of all Guinean health programmes has been severely disrupted by the Ebola epidemic and more recently the Covid pandemic.

The healthcare professionals in Conakry and its suburbs interviewed for this study noted that the current system for transporting samples for early diagnosis of children exposed to HIV was inefficient. The national programme vehicle that provided the shuttle service to the reference laboratory only travels when around twenty samples have been collected, forcing small health facilities that only follow a small number of HIV-exposed children to store these samples, sometimes for several months, before requesting for the national programme vehicle. Furthermore, doctors and midwives complained that they receive the results late, which compromises timely treatment initiation of HIV-positive children.

*"They talk about early diagnosis, but when you receive the results six months after the sample has been taken, it's no longer early!"* lamented a pediatrician

(B8, female, pediatrician)

In addition, regular shortages of reagents slow down laboratory analyses. Rather than stock-pile blood samples, health professionals prefer to suspend blood sampling. Parents are often told the results of their children's screening by social workers, particularly psychosocial assistants, some of whom are members of associations for people living with HIV. They found that waiting for the results increased the mothers' anxiety tenfold and added to the pressure:

*"Women are anxious to have their results, they call me on the phone, some come every day, they keep asking 'what is the result of my child?'* explains a psychosocial assistant

(C3, female, psychosocial assistant).

The long wait for results made it difficult for mothers to return to the health centre. While some went to the hospital as soon as they received a call from the medical staff, it is was not so easy for others. HIV remains a stigmatising disease. Many mothers living in polygamous households had not disclosed their HIV status with their family, or even to their husbands. They found it difficult to justify taking their child to hospital and paying for transport.

*"In our country, when a woman has given birth, we don't let her go out easily"* says a member of an association of PLHIV

(C1, female).

The psychosocial assistants noted that the longer it took for results to be delivered, the more difficult it was to contact mothers, who did not always have a personal telephone and sometimes moved house after giving birth.

The level of knowledge that mothers' had around early diagnosis varied. Some women knew nothing about the tests carried out during pregnancy and post-childbirth. They had not received the appropriate information and had not felt confident enough to question the doctor. Those who have taken part in research projects tended to be better informed. They remembered how anxious they were while waiting for results. Most reported that they would have been ready to wait on site for the time needed to obtain the result and minimise their anxiety. However, they were completely unfamiliar with the arrangements for transporting and analysing results on site, or in an external laboratory.

Most HIV-positive women knew that it was important to treat their children quickly if they tested HIV-positive. This had been communicated to them repeatedly by healthcare workers, and one paediatrician said that HIV-positive women were often in a hurry to get the drugs for their child if it tested HIV-positive.

## Views on the use of drones

**What the public and healthcare professionals knew about drones.** The level of knowledge about drones varied widely. Among the local population, some people, particularly students, were relatively well informed because they were connected to social networks and had an interest in current affairs. But the majority had a limited knowledge. The main sources of information were television and the internet. Many people discovered the use of drones through the American TV show "24", but also during the public protest in Conakry, where images were filmed by journalists, members of the opposition, or private individuals using drones. Health professionals and health authorities had a better understanding of drones, particularly their use in the medical field. The Rwandan blood bag transport programme is the best known. Several public health officials had witnessed flights during trips to Rwanda.

**Overall positive perceptions on the medical use of drones.** The use of drones in general, and specifically in the medical field, was perceived positively. In a turbulent political context where feelings of mistrust towards the incumbent government were widely shared at the time of data collection, the images filmed by the drones and retransmitted on television, or on the internet, revealed the scale of popular mobilisation and sometimes the abusive use of force during police repression. The use of drones was considered as a counter-power and as a means of accessing and sharing information.

*"It's good. People from outside can find out what's really going on in Guinea",* says a student

(F2, male, student).

Most of those interviewed felt that the medical use of drones met the needs of local communities in Guinea. Some perceived it as a way of solving the problems of traffic congestion in Conakry and minimising feelings of isolation for communities who resided outside of the capital. A number of people have described situations where loved ones had died due to a lack of rapid assistance or during ambulance transport because of traffic jams. Caregivers and community workers who already work with the NGO Solthis are most in favour of using drones to transport samples for early HIV testing of children exposed to HIV. They believed that the same-day delivery of the results would save time, limit the cost of transport for mothers and reduce the anxiety of waiting. It would also allow ARV treatment to be given immediately to HIV-positive children reducing the risk of mothers not returning to the health centre. However, some believed that the real emergency lies outside of Conakry.

*"If it's for Conakry, I don't really see the point. If it's for Kindia or Forécariah, or for other prefectures, if we could have drones, that would be extraordinary, because we can't have a laboratory for every site. At the moment, we don't have access to early screening for children there, whereas here, even if there are delays, we can have it",* explains a physician

(B6, male, medical doctor).

The use of drones for medical purposes, was also raising hopes of solving other public health problems, such as transporting blood bags in emergencies, as the head of a maternity unit explained:

*"In maternity wards, 80% of women die for lack of blood products. If your drone could transport bags of blood between the blood bank and here, it would be a great help. The day before*

*yesterday, a woman gave birth in a health centre. She suffered a haemorrhage and was transferred to us. We did the haemostasis, but there were no blood products. The parents went to get some. With all the traffic jams, by the time they got to the blood bank, collected the bags and came back, the woman was dead. It's common, it hurts me, I cry"*

(B7, male, medical doctor).

Healthcare professionals and members of associations for people living with HIV, see drones as a technological innovation that is part of the progress made over several decades to improve access to care, particularly in the fight against HIV. They perceived it as a further step forward, following on from rapid diagnostic tests, single-dose ARV drugs and the development of POCs for rapid results. Mothers who have experienced difficulties at the beginning of their care feel that medical progress was protecting them and their children.

The development of drone technology in the healthcare sector is also of interest to local health authorities in Guinea. They explained that it was part of the government's technical development strategy, which was supported by national government health officials who were interested for a range of reasons: to improve health indicators, particularly PMTCT, and achieve global health objectives; and out of national pride, to be at the forefront of technological innovation, following the example of countries such as Rwanda and Ghana.

*"The President of the Republic talks every day about new technologies, about the 4$^e$ industrial revolution, which we must not miss this time",* explains an adviser to the Ministry of Health

(A2, male).

The Ministry of Health was considering the future use of drones to deal with health emergencies and improve the decentralisation of the health system. The introduction of an initiative to transport samples by drone, as part of the early diagnosis of newborn babies exposed to HIV, was seen by those in charge as an opportunity to experiment with such a system.

*"At the Ministry of Health, we have some ideas about this. If it works with AIDS, we can use it to transport blood bags and reagents. If the experiment is successful, we can extend it to the whole country, for transport between the prefectures and the regions where we have set up laboratories. Before Ebola, we used to send everything to Conakry, but now we're giving the regions greater autonomy. If it works, we can add other opportunities to this project"* says an adviser to the Ministry of Health

(A4, male).

**The concerns raised by the use of drones.**   One of the primary fears is the hijacking or misuse of drones. Most people are familiar with the military use of drones, partly through TV shows, such as the famous "24 ", which seems to be the best-known reference, but also through current global events. In fact, just before the first period of data collection, in January 2020, the United States carried out a drone attack on a convoy in Iraq and killed an Iranian general. During our interviews, a number of people referred to this and to drones being utilised as weapons of war in the Sahel region of Africa. In this context, the rapid development and use of drones, was fuelling a number of fears, including that of civilian aircrafts being hijacked by foreign mercenaries or terrorists. The health and military authorities we met were also concerned about the security of drones so that they cannot be hijacked for criminal purposes.

*"If it's not secure, people can use it for another purpose. Killing people. You never know. The jihadists are on all fronts,"* worries a Defense Ministry employee

(A7, male).

Accidents involving drones, such as collisions with birds, human mishandling or bad weather, were also a cause for concern in a country where heavy rain and gusty winds are common during the winter. Some people were worried about the risk of human injury in the event of a fall on a person or vehicle. There were also concerns surrounding fear of contamination if someone were to come in to contact with a broken test tube with blood containing the AIDS virus. Caregivers and patients questioned the preservation of sample quality and the reliability of results, but expressed no concern about the risk of confidentiality being breached during air transport.

Some have expressed doubts about the relevance of introducing what they deem a costly and sophisticated technology, even though basic needs in terms of medicines and diagnostic tests are not being met. They were also not convinced that transporting samples and delays in reporting results were the main problems in paediatric HIV care.

*"We are constantly running out of screening tests and reagents for measuring viral load, especially paediatric ARVs. Some children are on bi-therapy because they don't have enough drugs. If we can diagnose children, but then we can't treat them, what progress has been made"*, says an NGO leader

(E1, male, medical doctor)

Furthermore, despite the high hopes for drones, officials from the Ministry of Health and the Civil Aviation Authority questioned the current capabilities of drones in terms of range and distance covered. They see these limitations as an obstacle to the development of regional programmes.

*"I had asked my colleague in Rwanda, and I realised that the distances between towns in Rwanda are very close, no more than 50 km. The situation in Guinea is different: the distances are considerable, especially in the forest regions"*, says a Health Ministry adviser

(A4, male).

**The need to regulate the use of drones.**   In recent years, the Guinean civil aviation authorities (AGAC) have noted an "explosion" of initiatives concerning the use of drones in Guinea, in the mining, agricultural, electrical, environmental, and now health sectors. They believe that a legislative framework needs to be implemented, and this is currently being developed to define the type of aircraft to be used, pilot qualifications and airworthiness requirements, particularly in urban areas. They plan to set up a pilot training school and envisage the production of drones in Guinea in the future. Those in charge said they were keen to regulate the use of drones, not to restrict them, but to ensure flight safety.

*"We're not against progress, on the contrary, we're up for it, it's the future. It's up to us to take the lead so we don't get left behind. We know the usefulness of drones, we're in favour of them, we can't be against technology. Before we had telephones, now we have smartphones. But there are downsides that need to be dealt with. We are the air marshals"*, explains a civil aviation official from Guinea

(A8, male).

For most of the people involved in the study, whether health professionals, managers or ordinary citizens, another prerequisite for the use of drones, particularly for transporting blood samples, was informing the public. In their view, the idea of potentially contaminated blood being flown overhead could lead to panic or even sabotage attempts. In their view, clear and wide-ranging communication is essential to reassure the public and prevent the spread of rumours. Most people referred to the rumours that were rife during the Ebola epidemic.

*"There was talk that the state was transporting the Ebola virus by drone to contaminate the population, and there was also talk of organ trafficking. I believed it myself",* adds a student

(F2, male).

## Discussion

### Alignment with public health priorities

The need to improve transport of samples to optimize diagnosis of HIV-positive children is a major concern for health authorities, particularly in regions outside the capital. The use of drones could accelerate the diagnosis of HIV-exposed children, enable early treatment of HIV-positive children and reduce mortality, which is known to be highest in the first two years, in the absence of adequate treatment [2].

The potential benefits of drones in improving access to healthcare for populations that are hindered by distance and lack of infrastructure have been widely demonstrated [10, 26–28]. At present, the technological limitations of drones restrict their capabilities in terms of distances covered and loads transported; furthermore, costs are still high [29–31]. As part of the AIR--POP project, the comparative cost-effectiveness analysis carried out by modelling showed that transporting blood samples by drone could be cost-effective in Conakry, in terms of life-years gained, compared with transporting them by motorbike [13]. Outside the capital, the situation is different: there are long distances between towns, and large areas of forest. However, the rapid progress currently being made in the development of drones suggests that in a few years, the performance of these devices will make possible and profitable what is not realistic at present [11, 32, 33].

For health authorities, the use of drones could meet other healthcare needs, such as the rapid transport of blood bags to address post-partum haemorrhages responsible for a significant portion of maternal mortality [34] or the supply of reagents and emergency medicines. Various studies have demonstrated, from a theoretical or operational point of view, the feasibility of transporting blood [10], vaccines [35], tuberculosis samples and medicines [11] and in Guinea, anti-epileptic drugs [36]. The design of a project using drones should take into account the possibilities of their mutualization for different health programmes.

Finally, the implementation of costly programmes, using cutting-edge technology, to the detriment of covering essential needs, raises concerns and misgivings among stakeholders involved in development programmes. These concerns go hand in hand with ethical issues raised about drones. Some authors stress the importance of not destabilizing healthcare systems [37], of avoiding technology supplanting infrastructure strengthening [38] and of developing "acceptable drones" rather than "drone acceptability" [39]. Sandvik warns against "technological utopia" and points out that the use of drones in the humanitarian field in Africa is not always without ambiguity. For example, coastal surveillance by drones can contribute to the rescue of migrants at sea, but they can also be used for border control [40].

## The ambivalent perception of drones

Overall, drones benefit from a relatively positive image in Conakry as they do in many other countries. Various studies report a general acceptance of the use of drones, particularly in the health sector. In the United States and Canada, their use in emergency medicine is considered relevant and reassuring by both professionals and the public [41, 42]. In Latin America and Asia, their use in the fight against Dengue fever is attracting considerable public support [27]. In Africa, studies of acceptability and experience show that populations are very receptive to the use of drones to optimise the fight against malaria [43] and transporting medical supplies [44].

Drones convey an image of modernity and technical progress. The attraction of these technologies, described as "humanitarian neophilia" [45] demonstrated by both the authorities and the general population, is a way of seeing technological innovations as tools for improving access to information or care. The introduction of drones into the arsenal of global and national health systems is a source of pride for national leaders, in a context where Guinea is often cited as one of the nations with the lowest health and development indicators [6, 34].

But perceptions of drones are also sometimes negatively linked to war. They were first used for military purposes, and the image of the "killer drone" is still fresh in peoples' minds, especially the version that is frequently portrayed in popular culture through films, television, and global news outlets. As drones are used in the Middle East and more recently in Central Europe—with the Ukrainian conflict—to spot terrorist targets and launch attacks against them, their noise and silhouette in the sky is associated with fear for the local population in these conflict zones, who regard any drone as a "killer drone" [46]. Mistrust of drones has led some countries, such as Bhutan, to ban their use [47]. Although there has been no direct experience of the military use of drones in Guinea, these representations, relayed by the wider international media, are fuelling the concerns of the Guinean people and the authorities.

Our interlocutors also expressed other concerns related to possible drone malfunctions, particularly in the event of a crash. On the other hand, the fears relating to invasion of privacy described in other countries [44, 48], were not mentioned.

## Acceptability linked to the political and health context

This ambivalence entails a risk of fluctuating acceptability depending on changes in the political and health policy context, both nationally and internationally. From a geopolitical point of view, the current situation in West Africa is fragile. It is marked by an increasing number of coups d'état, as well as attacks by various jihadist movements. If conflicts on Guinea's borders escalate, drones could be used as weapons of war, as is already the case in the Sahel. The perception of an imminent threat could then exacerbate the Guinean population's concerns and mistrust of drones and call into question their support for health programmes that use them peacefully. The sometimes questionable practices of counter-terrorism policies are contributing to public scepticism of healthcare providers [49].

On the health front, the outbreak of epidemics often generates a series of concerns fuelled by the spread of rumours on social networks. The 2014–2016 Ebola epidemic in West Africa was marked by a profusion of rumours that slowed down the implementation of measures to combat the epidemic [50]. During the Covid-19 epidemic in 2020, vaccination programmes had to contend with *rumours* spread via social networks [51]. The use of drones to help combat the Ebola epidemic were raised as early as 2015 in West Africa [52] and were implemented in the Democratic Republic of Congo in 2019 [53]. The Covid-19 epidemic has seen a proliferation of initiatives in Africa to deliver diagnostic tests by drones [54] or transportation of samples [55]. In Guinea, the people we spoke to reported rumours circulating during the Ebola

epidemic about the deliberate dissemination of the virus among the population, organised by the state through drones. There is every reason to fear that similar rumours could spread in the event of another epidemic. This concern could lead to the public rejecting any current and future drone programmes.

## The need for good information

In view of the risk of rumours and misinformation, clear communication prior to the implementation of drone programmes is essential. The interviews showed that a considerable number of people have limited knowledge about drones, which is corroborated by other studies in different contexts [56–58]. In the absence of information enabling an informed opinion to be developed, adherence to programmes is guided by the confidence of the population and healthcare stakeholders, in partnership with whom collaborative links have previously been forged [37]. However, this trust, without any real consent and without sufficient information, could be compromised in the event of events giving rise to mistrust of drones, as we have just explained.

The need for clear and comprehensive information is widely accepted among key stakeholders who are involved in the development of drone programmes. According to Knoblauch, the information should cover not only the intentions of the programme, but also the operation of the devices, their advantages and disadvantages, operational details (e.g. flight paths and times, products transported) and be accompanied by a visual presentation of the object [44]. Strategies involving informing the authorities to whom people refer have been shown to contribute to their support [43]. UNICEF has developed a communication model to inform the public when setting up a humanitarian drone corridor in Malawi. It includes flight demonstrations and group discussions aimed at demystifying drones and sparking questions and discussions [59]. Communication strategies must also be adapted to the diversity of the local community concerned. Shapira & Cauchard highlight the differences in perceptions of drones that can exist between social groups and advocate for the development of "tailor-made" communication campaigns, or even the design of drones, that take cultural differences into account [60]. Whatever the strategy adopted, informing the stakeholders and populations concerned by these experimental programmes is an essential prerequisite for fostering subsequent acceptance.

## Study limitations and strengths

The main limitation of the study is its prospective dimension. Views were gathered after only one test flight. Only a few of the people interviewed actually witnessed the flight, and most of them based their opinions on the use of drones in contexts unrelated to this experimental flight. However, it is clear that the acceptability of an innovation evolves over time [61]. The direction of this evolution—favorable or unfavorable—and the factors that will determine it, cannot be predicted. Our study therefore describes an initial state. It will therefore be necessary to conduct a complementary study once the device has been set up and is being used routinely, enabling us to assess the acceptability of drone use in a context of use.

Thus, despite the limitations of the study mentioned above, the strengths of our study lie in our attempt to gather the views of the various social actors who are likely to be affected by the use of drones. This research is part of the anthropological contribution to the analysis and understanding of the acceptability of health innovations. From a public health perspective, our analyses provide elements for optimizing health programs using drones and to maximise public acceptance. Our findings deserve to be shared and compared with those of other countries,

to take account of the factors influencing the acceptability of the use of drones in healthcare programmes, in a context where such programmes are growing rapidly in Africa.

## Conclusion

Drones are both feared and coveted objects. They arouse various expectations and misgivings. It is difficult to predict whether an innovation such as the use of drones to optimise the transport of healthcare products will ultimately be effectively adopted. Certain relatively recent technological innovations have enjoyed undeniable success. The widespread global use of mobile phones and then smartphones is a prime example of the appropriation of an innovation [62]. Support for healthcare programmes using drones will depend on the perceived relevance of the programmes and their ability to meet the needs and expectations of healthcare professionals and the general public.

The results of the study show that perceptions of drones are generally positive, despite concerns about their potential misuse. Their use to facilitate the early diagnosis of children exposed to HIV is gaining support from mothers, carers and health authorities. However, this consensus is fragile, knowledge on the subject is sometimes limited and points of view may vary depending on changes in the political and health policy context. It is essential to keep healthcare professionals and the general public well informed about the objectives of drone programmes as well as the technical aspects. Furthermore, support for the project depends on whether it is in line with the health priorities as perceived by healthcare professionals, the authorities and the local community. It is also essential to take these priorities into account in order to adapt the project to the needs identified by stakeholders—and not the other way round. Questions of ethics and equity must be carefully considered, to ensure that "technological utopia" does not prevail over the principles that guide programme design, and that the use of drones in humanitarian and development work does not join the "graveyard of innovations" of which the past is the crucible [23].

## Supporting information

**S1 Checklist. Inclusivity in global research.**
(DOCX)

**S2 Checklist. COREQ (COnsolidated criteria for REporting Qualitative research) checklist.**
(PDF)

## Acknowledgments

The authors acknowledge David Nelson who edited the final version.

## Author Contributions

**Conceptualization:** Gabrièle Laborde-Balen, Mohamed Cissé, Youssouf Koita, Bernard Taverne, Maxime Inghels, Guillaume Breton.

**Data curation:** Gabrièle Laborde-Balen, Oumou Hawa Diallo, Bernard Taverne.

**Formal analysis:** Gabrièle Laborde-Balen, Bernard Taverne, Guillaume Breton.

**Funding acquisition:** Guillaume Breton.

**Investigation:** Gabrièle Laborde-Balen, Oumou Hawa Diallo, Mohamed Cissé, Maxime Inghels, Guillaume Breton.

**Methodology:** Gabrièle Laborde-Balen, Guillaume Breton.

**Supervision:** Gabrièle Laborde-Balen, Mohamed Cissé, Guillaume Breton.

**Validation:** Gabrièle Laborde-Balen, Oumou Hawa Diallo, Mohamed Cissé, Youssouf Koita, Bernard Taverne, Maxime Inghels, Guillaume Breton.

**Writing – original draft:** Gabrièle Laborde-Balen, Oumou Hawa Diallo, Bernard Taverne.

**Writing – review & editing:** Gabrièle Laborde-Balen, Oumou Hawa Diallo, Mohamed Cissé, Youssouf Koita, Bernard Taverne, Maxime Inghels, Guillaume Breton.

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
