## [Decision Letter · Decision Letter 0]

7 Feb 2024

PGPH-D-23-02263

Using drones to improve care for HIV-exposed children in Conakry, Republic of Guinea: anthropological perspectives

Dear Dr. Laborde-Balen,

Thank you for submitting your manuscript to PLOS Global Public Health. After careful consideration, we feel that it has merit but does not fully meet PLOS Global Public Health’s publication criteria as it currently stands. Therefore, we invite you to submit a revised version of the manuscript that addresses the points raised during the review process.

EDITOR:

Please respond to all of the reviewers comments.

Please use the COREQ reporting guidelines for qualitative research for your manuscript.

Please edit the paper for language. Active voice is preferred and always place the individual before the disease e.g., people living with HIV instead of "HIV patient"

Additionally, I request you to kindly consider the following:

Theoretical framework: Kindly identify a single theoretical framework that best applies to your study and describe how your study fits the framework- rather than describing multiple frameworks.

Methods: Briefly describe the PMTCT program in Guinea to provide context to the study for the benefit of the global audience.

Who collects the blood samples from childrenhow much sample is collectedwhat procedure is followedare samples collected from all neonates or only from neonates born to mothers with HIV infection?At what time point after birth are they collectedHow are the samples routinely transported.Who loads the samples in the drones and deploys themwho receives the drone.

Routinely, what is the turnaround time (TAT) for lab test reports for HIV in neonates?

Please describe the intervention: what kind of drones were used. Who operated them? How were they deployed. What was their capacity. Who loads the samples and how many are loaded at a time. How many drones were deployed? What area was covered? What was the back-up plan if the drones malfunctioned/ did not reach their destination?

Data collection and study population:

First describe the study population in detail and then the methods used in data collection. How many languages did the data collectors (interviewers) master.

Line 194: it is unclear what is meant by participants were selected based on reasoned choice- do you mean purposively, if yes please say so?

Analysis method:- better written as ‘analytical plan’ or just ‘analysis’ instead. What were the roles of the researchers in the analysis. Was an interview guide prepared? Please provide as an annexure/ additional information.

Kindly provide examples of the coding framework/matrix used in the content analysis as well as a clearer more detailed description of the steps used in the analysis. How were the concepts grounded theory used in the analysis? This especially as we do not see the results of an inductive analysis with the development of a hypothesis which is expected in the grounded theory. Some clarification on this would help.

Line 221: Please replace to-ing and fro-ing with suitable alternatives (back and forth for example)

Ethical aspects: Please replace with Ethical concerns

Results:

Kindly provide demographic details of your study participants: sex, age, residence (rural urban), occupation/ role in the healthcare system, education or any other that can comprehensively describe your study population- may be presented in a table.

Please tag/ label the quotes with anonymized unique identifiers (ensuring confidentiality)- for e.g., participant 5 or participant 6. This is to ensure that not all quotes were taken from the same interview.

Please rephrase the themes and subthemes as they are too cryptic. They could benefit from a little more detail and context.

For example: (i) Overall positive perceptions- of what?

(ii) but also fears- this is incomplete and unclear what you are referring to here?

Kindly provide only relevant text in the form of quotes- For examples line 321 to 327 under “overall positive perceptions” may be shortened by providing only the essential part of the text that just conveys how the drones could save lives.

Discussion: this should be restricted to relevant findings and without speculations. Also, please use subheadings minimally in the discussion and only where relevant. The discussion is not for the presentation of themes but for the discussion of the themes that have been identified in your study. Therefore, as such, the discussion should keep in mind the objectives and the results and should be structured accordingly.

Also, there is limited discussion of how the drones can improve care for HIV infected children in Conakry.

Kindly provide a section on methodological issues in terms of trustworthiness and reflexivity and any other bias that may need to be addressed.

We look forward to receiving your revised manuscript.

Kind regards,

Rashmi Josephine Rodrigues, M.D., Ph.D.

Academic Editor

Journal Requirements:

2. In the online submission form, you indicated that "The data is sensitive because it is potentially identifying. Excerpts from anonymized interviews are included in the article. Data is available upon request to the authors."

3. Uploaded as supplementary information.

Additional Editor Comments (if provided):

Reviewers' comments:

Reviewer's Responses to Questions

**Comments to the Author**

1. Does this manuscript meet PLOS Global Public Health’s publication criteria? Is the manuscript technically sound, and do the data support the conclusions? The manuscript must describe methodologically and ethically rigorous research with conclusions that are appropriately drawn based on the data presented.

Reviewer #1: Yes

Reviewer #2: Yes

2. Has the statistical analysis been performed appropriately and rigorously?

Reviewer #1: Yes

Reviewer #2: No

3. Have the authors made all data underlying the findings in their manuscript fully available (please refer to the Data Availability Statement at the start of the manuscript PDF file)?

Reviewer #1: Yes

Reviewer #2: Yes

4. Is the manuscript presented in an intelligible fashion and written in standard English?

Reviewer #1: Yes

Reviewer #2: Yes

5. Review Comments to the Author

Reviewer #1: General comments:

-Overall, the article reads well and contributes timely and relevant knowledge to the research field.

-Some wide changes in language and grammar are however recommended. I suggest you review the whole document again for grammar, use of tenses, and general social scientific research language and reporting style. For example, use of first person is recommended when describing methods e.g., “We conducted a pilot study” rather than “A pilot study was conducted” (Line 33, 36). Also, it is recommended to use past tense/reported speech, and ensure consistency of tense throughout when reporting results (e.g., line 307, 309, 235 and onwards).

-Consider use of non-stigmatising language in HIV, for example ‘Living with HIV’ instead of ‘HIV-positive’ (e.g., line 281, 283 etc)-. See this link for details on the UNAIDS guidelines https://unaids-test.unaids.org/sites/default/files/unaids/contentassets/documents/unaidspublication/2011/JC2118_terminology-guidelines_en.pdf

-Review the document thoroughly for formatting issues e.g., spacing e.g., line 54, 74, 300, and consistency in formatting e.g., use of full stops including before or after citations or at the end of sentences, e.g., line 52, 88, 337, 410, 494, 528 and repeated words e.g., line 490, 502 etc.

Line by line comments:

Line 51: “a third of them reside”, consider ‘a third of whom reside’ or ‘a third of them resident’

Line 52 and across all citations- Check if the full stop before the parentheses is an acceptable format

Line 54- Remove spacing before parenthesis

Line 54, 88, 90 etc.- First mention of abbreviations should be written if full with abbreviation in parentheses

Line 56, 237 etc- ‘HIV-positive’ instead of ‘positive’

Line 60- Revise to past tense

Line 62- Are ‘people’ here synonymous to ‘children’?

Line 67- Consider “Would not be cost effective”

Line 81- Do you mean ‘present’?

Line 84 and elsewhere- Check for consistency of your citation format, is it in-text or using numbers?

Line 89- ‘Papua’?

Line 113- Clarify what you mean by ‘associations’. Are they associations of people living with HIV etc?

Line 39, 128-: The theories are well described at this point. However, there are questions on how these theories are related to the research questions. Are there particular variables from these theories that informed the study questions and research design? Did you consider using a relevant theoretical framework to frame the research findings and inform conclusions? For example, there are various frameworks/theories for assessing acceptability and/or innovation? You could look more into relevant frameworks like the “Unified Theory of Acceptance and Use of Technology” or the “Theoretical Framework of Acceptability”. These could be described more clearly in the methods section, in addition to the concepts presented, to help frame and explain your findings.

Line 132-133 The researcher, ‘Alter’ should be correctly cited at the first mention.

Line 178- Use of the word ‘ambivalence’ may be quite biased at this point since the reader has not interacted with the results yet

Line 187- Clarify what you mean by ‘ministerial structures’. Do you mean government ministries?

Line 192- It would be useful to briefly describe the researchers, e.g., gender, education background etc

Line 194- What kind of sampling strategy did you use? How did you identify and select participants? What do you mean by ‘reasoned choice’?

Line 202- Were all these individual interviews or were there group interviews in some cases? It would be good to clarify how many of these were open or semi-structured, and how many were observations?

Line 202-212- It would be clearer to use a table to present these demographics. Also having some descriptive demographics e.g., gender, age, institution, roles etc. would be more informative

Line 202, 219- Which language did you conduct interviews in? Did you translate any interviews? How else did you record your data e.g., for observations? It would be good to clarify this.

Line 220- Did you use any software for coding?

Line 234- Consider ‘Covid pandemic’

Line 249, 251 etc- Consider ‘HIV-exposed’, ‘HIV-infected’

Line 250- ‘Requesting for’

Line 253, 269 etc.- When reporting quotes, it is recommended you put them in context e.g., you could include interviewee type and outstanding demographics e.g., gender, role, etc. in parentheses

Line 288- ‘Limited knowledge’ instead of ‘more limited knowledge’.

Line 434- It would be useful to include the contexts mentioned here with relevant citations.

Line 441, 457- ‘could become closer’, ‘to reject any programme’- Rephrase to correct language.

Line 516, 520- ‘Certain’, ‘ability to’

Reviewer #2: Hi Gabriele,

Thanks for submitting the paper to PLOS GPH for review. The study looks interesting.

Few observations are as follows;

1) At background section;

- Why did you choose Conakry over other city like Nzérékoré?

- please mention what kind of POC device available for HIV testing at city of Conakry? (vide 93)

- you may think to shorten the paragraph on political instability as there is repetition. (vide 116)

2) At methodology section, this is not clear about the kind of framework used (theory by Olivier de Sardan, Alter, Akrich, Latour, Callon, Law and other theorists, Greenhalgh, Rogers, Louart) to fix up your approach (vide 128) and also explain a bit about your approach (Unified or mixed up) and its superiority.

3) At data collection, please explain observation (s?) (singular or plural) (vide 191)

4) At analysis, please mention 'recorded' in multi dialects (vide 219)

5)Please explain the logic for choosing grounded theory (vide 224)

6) At results section, please add participants' views on maintaining the confidentiality of subjects while collecting blood and testing vs. same by using Drone. (vide 234) (if data available)

7) This paper does not mention about the perceptions and/or fear and or surprise and or curiosity after seeing the drone landing on anyone's place or ground by risking the exposure of subject's identity.

8) You may add at conclusion section about the strategy to address the confidentiality of subject by seeing the drone as people may flock outside out of surprise or part of entertainment or out of curiosity or both.

9) The paper should also to mention the who is going to collect the blood from the subject before the drone flies off.

Please incorporate accordingly.

Thanks and Best wishes,

Braj

6. PLOS authors have the option to publish the peer review history of their article (what does this mean?). If published, this will include your full peer review and any attached files.

**Do you want your identity to be public for this peer review?** For information about this choice, including consent withdrawal, please see our Privacy Policy.

Reviewer #1: No

Reviewer #2: **Yes: **Dr. Brajaraj S Ghosh

---

## [Decision Letter · Decision Letter 1]

20 Aug 2024

PGPH-D-23-02263R1

Using drones to improve care for HIV-exposed children in Conakry, Republic of Guinea: anthropological perspectives

Dear Dr. Laborde-Balen,

Thank you for submitting your manuscript to PLOS Global Public Health. After careful consideration, we feel that it has merit but does not fully meet PLOS Global Public Health’s publication criteria as it currently stands. Therefore, we invite you to submit a revised version of the manuscript that addresses the points raised during the review process.

Kindly ensure that the reviewers concerns are addressed.

We look forward to receiving your revised manuscript.

Kind regards,

Rashmi Josephine Rodrigues, M.D., Ph.D.

Academic Editor

Journal Requirements:

Additional Editor Comments (if provided):

Reviewers' comments:

Reviewer's Responses to Questions

**Comments to the Author**

1. If the authors have adequately addressed your comments raised in a previous round of review and you feel that this manuscript is now acceptable for publication, you may indicate that here to bypass the “Comments to the Author” section, enter your conflict of interest statement in the “Confidential to Editor” section, and submit your "Accept" recommendation.

Reviewer #1: All comments have been addressed

Reviewer #2: All comments have been addressed

2. Does this manuscript meet PLOS Global Public Health’s publication criteria? Is the manuscript technically sound, and do the data support the conclusions? The manuscript must describe methodologically and ethically rigorous research with conclusions that are appropriately drawn based on the data presented.

Reviewer #1: Yes

Reviewer #2: Yes

3. Has the statistical analysis been performed appropriately and rigorously?

Reviewer #1: Yes

Reviewer #2: Yes

4. Have the authors made all data underlying the findings in their manuscript fully available (please refer to the Data Availability Statement at the start of the manuscript PDF file)?

Reviewer #1: Yes

Reviewer #2: Yes

5. Is the manuscript presented in an intelligible fashion and written in standard English?

Reviewer #1: Yes

Reviewer #2: Yes

6. Review Comments to the Author

Reviewer #1: I commend the authors for thoroughly reviewing the initially submitted manuscript draft. They have sufficiently addressed the comments raised in the initial review and presented a higher quality draft considering that English may not be their primary language which could have affected the initial expression of thoughts.

Generally, the authors should consider reviewing the results section for consistency of tenses used. Currently the authors use present continuous tense and past tenses interchangeably. Since they are reporting past findings, the past tense is recommended for reporting the results.

New comment:

The authors could consider highlighting the strengths, if any, of their study, not just the limitations.

Line by line comments:

Line 106: Remove ‘of’ in United Nations International Children's Emergency Fund

Line 244: Has a typographical error, “lassting”

Line 373: Add ‘to’ HIV-positive children

Line 512: “Raised by drones” sound like drones have raised the issue. Consider, “raised about drones”

Reviewer #2: Authors addressed all comments as desired. The revised article looks OK to me.

7. PLOS authors have the option to publish the peer review history of their article (what does this mean?). If published, this will include your full peer review and any attached files.

**Do you want your identity to be public for this peer review?** For information about this choice, including consent withdrawal, please see our Privacy Policy.

Reviewer #1: No

Reviewer #2: No

---

## [Decision Letter · Decision Letter 2]

25 Oct 2024

Using drones to improve care for HIV-exposed children in Conakry, Republic of Guinea: anthropological perspectives

PGPH-D-23-02263R2

Dear Dr Laborde-Balen,

We are pleased to inform you that your manuscript 'Using drones to improve care for HIV-exposed children in Conakry, Republic of Guinea: anthropological perspectives' has been provisionally accepted for publication in PLOS Global Public Health.

Best regards,

Julia Robinson

Executive Editor

Reviewer Comments (if any, and for reference):

Reviewer's Responses to Questions

**Comments to the Author**

1. If the authors have adequately addressed your comments raised in a previous round of review and you feel that this manuscript is now acceptable for publication, you may indicate that here to bypass the “Comments to the Author” section, enter your conflict of interest statement in the “Confidential to Editor” section, and submit your "Accept" recommendation.

Reviewer #1: All comments have been addressed

2. Does this manuscript meet PLOS Global Public Health’s publication criteria? Is the manuscript technically sound, and do the data support the conclusions? The manuscript must describe methodologically and ethically rigorous research with conclusions that are appropriately drawn based on the data presented.

Reviewer #1: Yes

3. Has the statistical analysis been performed appropriately and rigorously?

Reviewer #1: Yes

4. Have the authors made all data underlying the findings in their manuscript fully available (please refer to the Data Availability Statement at the start of the manuscript PDF file)?

Reviewer #1: Yes

5. Is the manuscript presented in an intelligible fashion and written in standard English?

Reviewer #1: Yes

6. Review Comments to the Author

Reviewer #1: The authors have adequately addressed the comments raised in the previous review. I congratulate the authors for their through review. The current manuscript is of a high quality and will add new knowledge to the field. A thorough proof reading of the manuscript is advised to remove any grammatical and typographical errors before publication.

For example:

Line292 caring for children HIV-positive = caring for HIV-positive children

323 hade not disclosed= had not disclosed

Line 393 seen drones= see drones

398 care feelthat= feel that

7. PLOS authors have the option to publish the peer review history of their article (what does this mean?). If published, this will include your full peer review and any attached files.

**Do you want your identity to be public for this peer review?** For information about this choice, including consent withdrawal, please see our Privacy Policy.

Reviewer #1: No
